# Cost–Benefit Analysis of Interventions to Mitigate the Monkeypox Virus

**DOI:** 10.3390/ijerph192113789

**Published:** 2022-10-23

**Authors:** Ali Mofleh ALSHAHRANI

**Affiliations:** Department of Clinical Pharmacy, College of Pharmacy, TU, KSA, P.O. Box 888, Haweiah 21974, Saudi Arabia; a.shahrani@tu.edu.sa or dr.alawi111@hotmail.com; Tel.: +966-555089203 or 996-0127272020 (ext. 2518)

**Keywords:** cost, benefit, monkeypox, virus, outbreak, pandemic

## Abstract

When a viral outbreak occurs, governments are obligated to protect their citizens from the diverse adverse effects of the disease. Health policymakers often have several interventions to consider based on the health of the population, as well as the cascading social and economic consequences of the possible mitigation strategies. The current outbreak of the monkeypox virus has elicited debate on the best mitigation strategy, especially given that most world economies are still recovering from the harsh economic effects of the COVID-19 pandemic. This paper sought to analyze the costs and benefits of three possible strategies and determine which option has the best health outcomes and positive economic effects. A case study of Jeddah was performed, whereby a model was simulated to determine the number of infections over 28 days based on one case of the monkeypox virus. Findings reveal that the vaccination provides the best intervention, as it effectively reduces the transmission rate and prevents loss of lives in the city. From the model, only three people were infected over the research period, while no deaths were reported. Although vaccination incurs a huge direct cost at the beginning, in the long run, it saves the economy from the disease’s financial burden in terms of productivity loss from work absenteeism and premature deaths.

## 1. Introduction

On 18 May 2022, a U.S. resident tested positive for monkeypox after returning to the U.S. from Canada [1]. As of 18 June 2022, the Centers for Disease Control and Prevention (CDC) reported 113 confirmed cases of monkeypox across multiple states. Days later, multiple monkeypox cases were reported in different countries that are not normally endemic to the virus. As of 22 June 2022, more than 40 countries that are not endemic to the monkeypox virus have reported viral disease outbreaks, as confirmed cases exceed 3000 [2]. What makes the current outbreak more concerning is the rapid, continuing spread into new nations and continents, and the risk of further sustained transmission into vulnerable populations, including people that are immunocompro-mised, pregnant women, and children.

Monkeypox is a rare disease caused by infection with the monkeypox virus. It was first documented in humans in the 1970s, and outbreaks have been reported in many countries, with most cases restricted to endemic areas [3]. Symptoms of an infected person include rashes, fever, headaches, muscle ache, swelling, and back pain as re-ported by the WHO [4]. Usually, the infections last two to four weeks. The monkeypox virus is typically endemic to Central and West African countries such as the Demo-cratic Republic of the Congo (which has the highest infection rate), Cameroon, Central African Republic, Cote d’Ivoire, Liberia, Gabon, Republic of the Congo, Nigeria, and Sierra Leone [3]. Although epidemiological investigations are still ongoing, reported cases thus far have not established travel links to endemic areas. Consequently, the unusually high number of people infected with monkeypox outside of Africa with no travel links to the region has created a sense of panic that the virus is now spreading globally. Cessation of smallpox vaccination programs, encroachment of humans into forested areas, and growing international mobility are suspected to be the main reason contributing to the current outbreak. So far, 72 deaths have been reported in eight countries [5]. However, these deaths all occurred in Africa, and non-endemic nations are yet to record fatalities from the virus.

Transmission of the monkeypox virus occurs through three main channels. First, an infection can occur after close contact with the skin lesions of an infected person [6]. Secondly, monkeypox spreads after a person/animal is exposed to large respiratory droplets of the infected person/animal during face-to-face contact, or exposure to oral fluids of an infected individual during intimate sexual contact [7]. Finally, as with the case of the infected UK health worker [8], the monkeypox virus can also be transmitted through contaminated fomites. 

While monkeypox and Covid-19 both spread through respiratory droplets during close face-to-face interaction, the latter is extremely infectious and only requires tiny droplets to spread. Despite being transmitted through respiratory secretions, monkey-pox is not a respiratory virus [9]. Instead, it predominantly spreads through direct (usually prolonged) contact with monkeypox rash, scabs, or body fluids from someone infected.

Many respiratory infectious diseases are transmitted through close per-son-to-person contacts. In their study, Leung et al. determined that the number of re-ported contacts and duration of interaction were influenced by age (inversely), house-hold size (direct), education (direct), and income level (direct). In their study, Leung et al. concluded that a person interacts with a mean of 8 contacts per day [10]. Indeed, this figure is ideal for the study given that the serological survey was conducted in Hong Kong, a hotspot for emerging infectious diseases due to its immense population density and high connectivity in the worldwide air-transportation network. During the duration of the study (2015), Hong Kong’s population was estimated to be 7.291 mil-lion. Moreover, Leung et al. considered respiratory viruses such as pertussis that can also be spread through face-to-face interactions or droplets deposited on contaminated surfaces [10]. In another study, Eams et al. investigated dynamic social contact pat-terns as a measure to explain the spread of the HIN1 virus. Like Leung et al., their findings showed that physical social contact patterns vary across age groups in an inverse version, with adults aged 65 years and above showing an average of 3 persons per day, those in the 19-64 years of age having an average of 5 physical contacts per day, and children of the school going age (5–18 years) having an average of 11 contacts per day [11]. Similarly, Dodd et al. age- and sex-specific social contact patterns and incidence of mycobacterium tuberculosis infection. The findings of their study showed that on average, adults in high density environments make about 10 social contacts on a single day [12]. Further, Johnstone-Robertson et al. designed a study to investigate social mixing patterns within a South African township community and the implications they have for respiratory disease transmission and control. The authors found that on average, adults make an average of 12 contact per day [13]. For this study, contact will comprise physical interaction between people, close and prolonged face-to-face contact, sexual contact, touching fabrics, objects, and surfaces that have been used by someone.

During past years, viral outbreaks caused devastating direct and indirect effects on the affected communities. The 1918 influenza pandemic is the most severe in mod-ern history [14]. Approximately 500 million people, constituting one-third of the world’s population at the time, were infected by the H1N1 virus [15]. Consequently, at least 50 million people worldwide died from the virus, with about 675,000 occurring in the United States [16]. What was especially concerning regarding this pandemic was the high mortality in healthy people, including those aged between 20–40 years [17]. The pandemic occurred in three waves, with the second and third waves claiming the most lives. India is believed to have suffered at least 12.5 million deaths during the pandemic [15].

Further, beyond the high death toll, the full impact of the 1918–1919 pandemic was realized about 60 years later. A study examining epidemiological data on individuals born in 1919, who were newborns or second- or third-trimester fetuses during the height of the pandemic, revealed that these individuals had approximately 25% more heart disease after age 60 [18]. Besides, this population exhibited an increased risk of diabetes in comparison with a similar population from a different era, including those who were older infants during the pandemic.

Considering that influenza was particularly deadly for young adults without pre-existing conditions, its economic impact was severe in comparison with other out-breaks that mostly affect young and aged citizens [19]. The sharp fall in economic activity coupled with heightened inflation resulted in massive declines in the real returns on stocks and short-term government bonds [20]. For instance, economies experiencing an average death rate of 2 percent saw real stock returns drop by 26 percentage points [21,22]. Garrett used historical newspaper reports to gauge the effects of the pandemic on US businesses. In the short term, businesses such as grocery stores and merchants/department stores report severe falls in consumer demand and labor shortages [23]. Barro, Ursúa, and Weng found that the Spanish flu reduced real GDP per capita by around 6 percent in the typical country over the period 1918–21 [24]. Correia, Luck, and Verner found that Spanish flu reduced US manufacturing output by 18 percent [25].

Correia, Luck, and Verner also emphasize the important link between government containment measures and economic outcomes [25]. Theoretically, the economic effects of containment measures could be positive or negative. While reviewing the im-pact of Covid-19 containment measures, the International Monetary Fund (IMF) concludes that containment measures have a significant impact on economic activity, equivalent to a loss of about 15 percent in industrial production over 30 days following their implementation [26]. Workplace absenteeism and reduction in productivity slow down across various industries [27]. Social distancing, self-isolation, and travel re-strictions during the peak of the Covid-19 pandemic caused reduced workforce and job losses across all economic sectors. However, such containment measures prevent the long-term loss of income caused by premature deaths [28]. After studying the spread and impact of the Severe Acute Respiratory Syndrome (SARS) epidemic in Toronto, Gupta, Moyer, and Stern explain that quarantine is effective in containing newly emerging infectious diseases, while cost-saving when compared to not implementing a widespread containment mechanism [29,30]. The SARS epidemic in Toronto resulted in relatively few infections and deaths. Had the virus not been contained, the re-search’s simulation model indicated that it could have spread throughout Canada as well as the rest of the world with devastating economic effects.

Although the risk to the general public remains low [3], close contact should be avoided with those suspected of the infection. These include those with skin or genital lesions, as well as sick or dead animals. CDC also urges those displaying symptoms of the virus, such as an unexplained skin rash or lesions, to avoid contact with others and reach out to their healthcare provider for guidance. Moreover, the WHO recommends promptly isolating suspected or confirmed cases in a single room with adequate ventilation, a dedicated bathroom, and staff until all lesions have resolved, the scabs have fallen off, and a fresh layer of intact skin has formed. One isolation strategy that has proved effective is quarantine, where a person suspected or found to have a disease is separated from others to prevent the spread of the virus. As Wang et al. explain, quarantine strategies effectively reduce the infection rate and delay new infections over time [6]. Consequently, countries such as Belgium, the U.K., and Germany have already introduced mandatory 21-day quarantine for patients confirmed to have been infected by the monkeypox virus [7,31]. However, critics such as Joe Biden, the U.S President, claim that quarantines are not feasible to address the ongoing monk pox pandemic.

Vaccination against smallpox has proved to be cross-protective against monkey-pox. According to the WHO, the smallpox vaccine is 85% effective in preventing monkeypox [5]. However, any immunity from smallpox vaccination is currently limited only to older citizens (minimum age varies among countries 42–50) since smallpox vaccination programs were stopped after the eradication of smallpox [32]. In addition, protection for those vaccinated may have waned over time. In 2019, the U.S. Food and Drug Administration (FDA) recently approved a live-attenuated vaccine, trademarked as JYNNEOS, that protects against infection of smallpox and monkeypox in adults 18 years of age or older [8].

Economic evaluation of health and health care has become an increasingly common component in health policy and planning. Cost-effectiveness analysis (CEA) formally assesses trade-offs involving benefits, harms, and costs inherent in alternative options. Such analysis explicitly quantifies the relative costs and benefits of alternative interventions [33]. Hence, this illuminates the potential trade-offs and informs discussions of whether a particular intervention (over an alternative) is worth its additional health gain. Cost-effectiveness analysis (CEA) has been used as a tool for addressing issues of efficiency in the allocation of scarce health resources, providing as it does a method for comparing the relative costs as well as health gains of different (and often competing) health interventions [9]. Net cost is calculated by subtracting the averted medical and productivity costs from the expenses of the intervention. This study will examine the costs and health outcomes of implementing quarantines and vaccinations to curb the new spread of the monkeypox virus [34].

## 2. Research Method

### 2.1. Research Objective

This study seeks to compare the costs and health outcomes of implementing quarantines or vaccinating people to mitigate the spread of the monkeypox virus. 

### 2.2. Research Questions

1. What is the cost of not adopting any intervention to curb the spread of the monkeypox?

2. What is the net cost incurred when quarantining patients with the monkeypox virus?

3. What is the net cost incurred to vaccinate patients with the monkeypox virus?

### 2.3. Research Methodology

This research will use a case study of Jeddah, a Saudi Arabian port city on the Red Sea. The study will be simulated to run in 28 days and will target the city’s entire population of 4.8 million people. The research will assume that one person, patient zero, contracts the virus and spreads it to the rest of the city. First, a model will be used to simulate the spread and effects of the monkeypox virus when an intervention is not used. Subsequently, the economic implications of the viral infection will be calculated over the study’s period. This will aggregate the medical costs of treating those infected and the productivity loss from both work absenteeism and mortality. Afterward, a quarantine facility will be introduced to isolate those infected by the virus. The costs of the quarantine and the medical expenses will be compared to the economic benefits of this intervention. Finally, the community will be vaccinated with the smallpox vaccine, which has proved to be 85% effective in preventing monkeypox. The cost of the vaccination will then be compared to the financial gain of this intervention.

## 3. Results

### 3.1. The Cost of Not Implementing Any Intervention

The government has an option to not implement any intervention and instead allow the virus to spread freely among the population. Hence, a model will be used to simulate the infection rate in the city and determine how many people will have contracted the virus within 28 days. In addition, the monkeypox infection rate will be used to determine how many of the infected people will die from the disease and the economic burden of their deaths on the community. 

The transmission rate of a virus is influenced by various factors, the most important being the rate of contact between infected and susceptible individuals. Using informed estimates and data about the transmission rate of the monkeypox virus, Jeddah’s population density, the number of contacts a typical person makes in one day, and the incubation period of the monkeypox virus, the spread of the virus throughout the population will be modeled. This study will use findings by Jezek et al., who sought to determine the probability of becoming ill following infection from a known human source [10]. From the 2278 people who had close contact with 245 monkeypox patients infected from an animal source, 93 fell ill and were presumed to have been infected from the known human source, albeit from different generations. Hence, there was an overall 3% probability of becoming ill following infection from a known human source. 

The number of contacts a typical person comes in close contact with on any given day varies across individuals and locations. In a study to determine the spread of respiratory infections across a population, Leung et al., concluded that people interact with an average of eight contacts per day [11]. Hence, on the first day, patient zero will transmit the virus to 3% of the people they interact with. Therefore, the number of infections on the first day will be calculated as:Number of new infections (nni) = transmission rate (tr) × number of contacts per day (cpd),(1)
Number of new infections on the first day (nni) = 3% × 8 = 0.24,(2)
Number of total infections (nti) = previous infections (pi) + transmission rate (tr) × number of total contacts per day (cpd),(3)
nti for the first day = pi + (tr × (cpd × nni)),(4)
nti = 1 + (0.03 × (8 × 1)) = 1.24,(5)
Number of total infections on the 2nd day (nti) = 1.24 + (0.03 × 8 × 1.24) = 1.5376(6)

Below is a summary of the infections during the 28-day window.

As such, by the end of the 28th day, **413 people** would have contracted the virus. Since no intervention will be taken, there are no direct expenses incurred. 

#### 3.1.1. Cost of Mortality

Researchers have observed that the monkeypox virus in non-endemic regions is less lethal than the strain observed in West Africa. Indeed, this is accurate since all 72 deaths so far have been reported from eight countries in Africa, and no non-endemic country has recorded fatalities from the virus [5] However, researchers still assign a case fatality ratio (CFR) of 1% for non-endemic countries [5]. Hence, 1% of the 413 people will succumb to the disease. 

0.01 × 413 = 4 deaths within 28 days;

In 2021, the GDP per capita in Saudi Arabia was USD 23,310. Therefore, the monthly wage of workers in RSA = USD 23,310/12 (months) working days;

The monthly wage of workers in RSA = USD 1942.5;

The daily wage of workers = USD 1942.5/21 (average number of working days in a month) = USD 92.5 [33,35].

The median age in Saudi Arabia is 31.8 years, while the life expectancy is 75.13 years [12]. Nonetheless, the official retirement age is 60 years. 

Hence, 28.2 years of productivity will be lost due to death. This amounts to 28.2 × USD 23,310 (GDP per capita) × 4 = USD 2,629,368 lost from the economy.

#### 3.1.2. Cost of Work Absenteeism

Most monkeypox patients heal and recover in 2 to 4 weeks. Hence, the 413 people will be in isolation for 28 days. The productivity loss incurred is calculated as:

The daily wage of workers × isolation days × number of patients;

USD 92.5 × 28 × 413 = USD 1,069,670;

Total loss to the economy from not taking any intervention = cost of mortality + cost of work absenteeism = USD 2,629,368 + USD 1,069,670 = USD 3,699,033.

### 3.2. The Net Cost of Quarantine

Quarantine effectively reduces the infection rate and delays new infections over time. In his research to determine the impact of quarantine on COVID-19 on infections, Marshall concludes that the average reduction in the infection rate is 24.9% [13]. Hence, the infection rate will reduce by 0.747%. Hence, the new infection rate will be:

3% − 0.747% = 2.253%

The number of infections for the 28 days based on the new infection rate will be simulated as:

### 3.3. Direct Costs

The direct cost of quarantine comprises the accommodation and treatment expenses [34]. During the COVID-19 pandemic, Saudi Airlines announced quarantine packages for various hotels in Riyadh, Jeddah, Medina, and Dammam. The quarantine facility in Riyadh is available at (USD 779) 2920 Saudi Riyals (SAR) for three-star hotels, (USD 1323) SAR 4965 for four-star, (USD 1833) SAR 6880 for five-star (Hilton Hotel), and SAR (USD 2063) 7744 for five-star (Lemeridian Hotel) (Syed, 2021). It is important to note that these costs cover the entire quarantine period. Thus, the average of the airlines’ quarantine packages is:

(USD 779 + USD 1323 + USD 1833 + USD 2063)/4;

Average cost of quarantine per room = USD 1500;

Cost of 104 rooms for 14 days of quarantine = USD 1500 × 104 = $156,000.

There are no treatments specifically for monkeypox virus infections [3]. For the majority of the patients, the signs and symptoms of monkeypox will resolve on their own, without any treatment. Plenty of fluids, good nutrition, and rest help the patients recover without medication. However, the risk of serious illness may be greater in pregnant women, children, and those with weaker immune systems. Monkeypox and smallpox viruses are genetically similar, which means that antiviral drugs and vaccines developed to protect against smallpox may be used to prevent and treat monkeypox virus infections [3]. Antivirals, such as tecovirimat (TPOXX), may be recommended for people who are more likely to get severely ill, such as patients with weakened immune systems. For smallpox, the Mayo Clinic recommends that adults should take 600 milligrams (mg) (3 capsules) twice a day for 14 days. On average, a dose of TPOXX costs USD 50 (SAR 188). Assuming that all the 5 patients will need medication, the treatment cost will be:

Cost of TPOXX = 104 × USD 50 = USD 5200;

Total direct costs = cost of accommodation + cost of medication = USD 156,000 + 5200 = USD 161,200.

### 3.4. Indirect Costs

#### 3.4.1. Cost of Mortality

Given that 1% of the 104 people will succumb to the disease,

0.01 × 104 = 1 death within 28 days.

Hence, 28.2 years of productivity will be lost due to death. This amounts to 28.2 × USD 23,310 (GDP per capita) = USD 657,342 lost from the economy.

#### 3.4.2. Cost of Work Absenteeism

The productivity loss incurred is calculated as:

The daily wage of workers × isolation days × number of patients;

USD 92.5 × 28 × 104 = USD 269,360;

Total loss to the economy from not taking any intervention = cost of mortality + cost of work absenteeism = USD 657,342 + USD 269,360 = 926,702;

The Total Cost of Quarantine = USD 161,200 (direct costs) + USD 926,702 (indirect costs) = USD 1,087,902.

### 3.5. The Net Cost of Vaccination

#### 3.5.1. Direct Costs

The direct cost of vaccination comprises the cost of purchasing the vaccine, as well as the budget to cater for the logistics involved in vaccinating the 4.8 million people. As part of the voluntary vaccination program, the government will announce that mandatory smallpox vaccinations would be available for free. However, CDC estimates that smallpox screening and vaccination efforts would cost the government USD 5 to USD 10 per patient [14]. Hence, this study will use USD 7.5 as the average cost of vaccinating each citizen. Therefore, assuming that the government vaccinates 100% of the population, the direct cost of this intervention will be:

USD 7.5 × 4.8 = USD 36 million.

#### 3.5.2. Indirect Costs

The smallpox vaccine has been shown to offer 85 percent of protection against monkeypox [3]. Hence, the infection rate will reduce by 0.0255, making the new infection rate 0.45%. The number of infections for the 28 days based on the new infection rate will be simulated as:

Cost of Mortality

Given that 1% of the 3 people will succumb to the disease,

0.01 × 3 = 0 deaths within 28 days.

Cost of Work Absenteeism

The productivity loss incurred is calculated as:

The daily wage of workers × isolation days × number of patients;

USD 92.5 × 28 × 3 = USD 7770;

Total loss to the economy from not taking any intervention = cost of mortality + cost of work absenteeism = USD 7770.

The Total Cost of Quarantine = USD 36,000,000 (direct costs) + USD 7770 (indirect costs) = USD 36,007,770.

## 4. Discussion

The health ministries can opt to implement either of the three strategies to curb the spread of the monkeypox virus. Each of these choices has different costs and benefits, as illustrated below.

Table 1, Table 2, Table 3 and Table 4 provide summary details of the simulation and the cost implications. Specifically, Table 1 is the simulated transmission rate assuming no government intervention measure is put in place, Table 2 presents a summary of the simulated transmission rate after the quarantine, while Table 3 and Table 4 summarize the transmission rate after vaccination and a comparison of the interventions respectively. As shown in Table 4, no expenditure would be incurred from not taking any measures to prevent the spread of the monkeypox virus. However, 413 people would contract the virus within 28 days, causing 4 deaths. In the long run, the isolation and deaths would adversely affect the nation’s GDP at USD 3,699,033. On the other hand, the direct costs of quarantine would be USD 161,200. Quarantine would help reduce the transmission rate of the virus, effectively minimizing the infections to only 5 within 28 days and a single death. The isolation of infected people and the mortality cost of the single life would amount to USD 926,702. Hence, the total cost of this intention would be 1,087,902. Finally, the government would spend USD 36 million to vaccinate the 4.8 million residents of Jeddah. Vaccinations would significantly reduce the infections to only three within 28 days, and no deaths would occur. Similarly, the productivity loss would be decreased to USD 7770.

The above findings show an inverse relationship between the direct and indirect costs among the three interventions. While not taking any measures would not incur any cost at the beginning, the long-term economic impact on the city would be immense. The number of infections and thereby deaths is increasing by the day, thus, causing a huge disease burden to the community. In the long run, this would cause a city pandemic, which will attract huge financial implications. On the other hand, quarantine attracts moderate direct costs and significantly reduces the number of infections and fatalities. However, the mortality cost and productivity loss are also huge and increase with time. While vaccination incurs a huge direct cost at the beginning, it effectively reduces the transmission rate and prevents the loss of lives in the city. This saves a huge amount of finances that would otherwise be lost due to isolation or death. Furthermore, vaccination is a one-off cost that will provide cost savings for many years, as it will ensure the community remains healthy and productive.

## 5. Conclusion

During a healthcare pandemic, the government is responsible for protecting its citizens from the diverse adverse effects of a disease outbreak. Albeit, at these times of unprecedented public health crisis, policymakers are required to respond swiftly and effectively while considering the cascading social and economic consequences of the available mitigation strategies. Health ministries can either choose to overlook the current wave of the monkeypox virus, introduce quarantines to isolate the affected or suspected cases, or vaccinate the population with the smallpox vaccine. The cost–benefit analysis of these interventions shows that the vaccination is the most rational option, as it would effectively limit the spread of the virus while protecting the nation’s productivity. As such, governments worldwide should allocate funds to support mass vaccination programs to protect their citizens from the health, social, and economic consequences of the monkeypox virus.

## Figures and Tables

**Table 1 ijerph-19-13789-t001:** Simulated transmission rate when no intervention is implemented.

Day	Transmission Rate	Default Contacts	Previous Infections	Total Contacts	New Infections	Number of Total Infections
**1**	0.03	8	1	8	0.24	1.24
**2**	0.03	8	1.24	9.92	0.2976	1.5376
**3**	0.03	8	1.5376	12.3008	0.369024	1.906624
**4**	0.03	8	1.906624	15.252992	0.45758976	2.36421376
**5**	0.03	8	2.36421376	18.91371008	0.567411302	2.931625062
**6**	0.03	8	2.931625062	23.4530005	0.703590015	3.635215077
**7**	0.03	8	3.635215077	29.08172062	0.872451619	4.507666696
**8**	0.03	8	4.507666696	36.06133357	1.081840007	5.589506703
**9**	0.03	8	5.589506703	44.71605362	1.341481609	6.930988312
**10**	0.03	8	6.930988312	55.44790649	1.663437195	8.594425506
**11**	0.03	8	8.594425506	68.75540405	2.062662122	10.65708763
**12**	0.03	8	10.65708763	85.25670102	2.557701031	13.21478866
**13**	0.03	8	13.21478866	105.7183093	3.171549278	16.38633794
**14**	0.03	8	16.38633794	131.0907035	3.932721105	20.31905904
**15**	0.03	8	20.31905904	162.5524723	4.87657417	25.19563321
**16**	0.03	8	25.19563321	201.5650657	6.046951971	31.24258518
**17**	0.03	8	31.24258518	249.9406815	7.498220444	38.74080563
**18**	0.03	8	38.74080563	309.926445	9.29779335	48.03859898
**19**	0.03	8	48.03859898	384.3087918	11.52926375	59.56786273
**20**	0.03	8	59.56786273	476.5429018	14.29628706	73.86414979
**21**	0.03	8	73.86414979	590.9131983	17.72739595	91.59154574
**22**	0.03	8	91.59154574	732.7323659	21.98197098	113.5735167
**23**	0.03	8	113.5735167	908.5881337	27.25764401	140.8311607
**24**	0.03	8	140.8311607	1126.649286	33.79947857	174.6306393
**25**	0.03	8	174.6306393	1397.045114	41.91135343	216.5419927
**26**	0.03	8	216.5419927	1732.335942	51.97007825	268.512071
**27**	0.03	8	268.512071	2148.096568	64.44289704	332.954968
**28**	0.03	8	332.954968	2663.639744	79.90919232	412.8641603

**Table 2 ijerph-19-13789-t002:** Transmission rate after quarantine.

Day	Transmission Rate	Default Contacts	Previous Infections	Total Contacts	New Infections	Number of Total Infections
**1**	0.02253	8	1	8	0.18024	1.18024
**2**	0.02253	8	1.18024	9.44192	0.212726458	1.392966458
**3**	0.02253	8	1.392966458	11.14373166	0.251068274	1.644034732
**4**	0.02253	8	1.644034732	13.15227786	0.29632082	1.940355552
**5**	0.02253	8	1.940355552	15.52284442	0.349729685	2.290085237
**6**	0.02253	8	2.290085237	18.32068189	0.412764963	2.7028502
**7**	0.02253	8	2.7028502	21.6228016	0.48716172	3.19001192
**8**	0.02253	8	3.19001192	25.52009536	0.574967748	3.764979668
**9**	0.02253	8	3.764979668	30.11983735	0.678599935	4.443579604
**10**	0.02253	8	4.443579604	35.54863683	0.800910788	5.244490391
**11**	0.02253	8	5.244490391	41.95592313	0.945266948	6.189757339
**12**	0.02253	8	6.189757339	49.51805872	1.115641863	7.305399202
**13**	0.02253	8	7.305399202	58.44319362	1.316725152	8.622124355
**14**	0.02253	8	8.622124355	68.97699484	1.554051694	10.17617605
**15**	0.02253	8	10.17617605	81.40940839	1.834153971	12.01033002
**16**	0.02253	8	12.01033002	96.08264015	2.164741883	14.1750719
**17**	0.02253	8	14.1750719	113.4005752	2.55491496	16.72998686
**18**	0.02253	8	16.72998686	133.8398949	3.015412832	19.74539969
**19**	0.02253	8	19.74539969	157.9631975	3.558910841	23.30431053
**20**	0.02253	8	23.30431053	186.4344843	4.200368931	27.50467946
**21**	0.02253	8	27.50467946	220.0374357	4.957443427	32.46212289
**22**	0.02253	8	32.46212289	259.6969831	5.85097303	38.31309592
**23**	0.02253	8	38.31309592	306.5047674	6.905552409	45.21864833
**24**	0.02253	8	45.21864833	361.7491866	8.150209175	53.3688575
**25**	0.02253	8	53.3688575	426.95086	9.619202877	62.98806038
**26**	0.02253	8	62.98806038	503.9044831	11.352968	74.34102838
**27**	0.02253	8	74.34102838	594.7282271	13.39922696	87.74025534
**28**	0.02253	8	87.74025534	701.9220427	15.81430362	103.554559

**Table 3 ijerph-19-13789-t003:** Transmission rate after vaccination.

Day	Transmission Rate	Default Contacts	Previous Infections	Total Contacts	New Infections	Number of Total Infections
**1**	0.0045	8	1	8	0.036	1.036
**2**	0.0045	8	1.036	8.288	0.037296	1.073296
**3**	0.0045	8	1.073296	8.586368	0.038638656	1.111934656
**4**	0.0045	8	1.111934656	8.895477248	0.040029648	1.151964304
**5**	0.0045	8	1.151964304	9.215714429	0.041470715	1.193435019
**6**	0.0045	8	1.193435019	9.547480148	0.042963661	1.236398679
**7**	0.0045	8	1.236398679	9.891189434	0.044510352	1.280909032
**8**	0.0045	8	1.280909032	10.24727225	0.046112725	1.327021757
**9**	0.0045	8	1.327021757	10.61617405	0.047772783	1.37479454
**10**	0.0045	8	1.37479454	10.99835632	0.049492603	1.424287143
**11**	0.0045	8	1.424287143	11.39429715	0.051274337	1.475561481
**12**	0.0045	8	1.475561481	11.80449185	0.053120213	1.528681694
**13**	0.0045	8	1.528681694	12.22945355	0.055032541	1.583714235
**14**	0.0045	8	1.583714235	12.66971388	0.057013712	1.640727947
**15**	0.0045	8	1.640727947	13.12582358	0.059066206	1.699794154
**16**	0.0045	8	1.699794154	13.59835323	0.06119259	1.760986743
**17**	0.0045	8	1.760986743	14.08789394	0.063395523	1.824382266
**18**	0.0045	8	1.824382266	14.59505813	0.065677762	1.890060027
**19**	0.0045	8	1.890060027	15.12048022	0.068042161	1.958102188
**20**	0.0045	8	1.958102188	15.66481751	0.070491679	2.028593867
**21**	0.0045	8	2.028593867	16.22875094	0.073029379	2.101623246
**22**	0.0045	8	2.101623246	16.81298597	0.075658437	2.177281683
**23**	0.0045	8	2.177281683	17.41825347	0.078382141	2.255663824
**24**	0.0045	8	2.255663824	18.04531059	0.081203898	2.336867721
**25**	0.0045	8	2.336867721	18.69494177	0.084127238	2.420994959
**26**	0.0045	8	2.420994959	19.36795968	0.087155819	2.508150778
**27**	0.0045	8	2.508150778	20.06520622	0.090293428	2.598444206
**28**	0.0045	8	2.598444206	20.78755365	0.093543991	2.691988197

**Table 4 ijerph-19-13789-t004:** Comparison of the interventions.

Intervention	Direct Costs (USD)	Infections (Mortality)	Mortality Cost	Productivity Loss	Indirect Cost	Total Costs
No Intervention	USD 0	413 (4)	USD 2,629,368	USD 1,069,670	USD 3,699,033	USD 3,699,033
Quarantine	USD 161,200	5 (1)	USD 657,342	USD 269,360	USD 926,702	1,087,902
Vaccination	USD 36,000,000	3 (0)	USD 0	USD 7770	USD 7770	USD 36,007,770

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
