# Peer review of "Cost–Benefit Analysis of Interventions to Mitigate the Monkeypox Virus"

_ijerph, 2022, doi:10.3390/ijerph192113789_

Round 1

Reviewer 1 Report

In this paper entitled “Cost-Benefit Analysis of Interventions to Mitigate the Monkey-pox Virus”, the authors described the current outbreak of monkeypox and discussed the cost-benefit outcome of different interventions. They compared the outcomes of monkeypox infection without interventions, with the intervention of quarantine facility or vaccination. By running a 28-day simulation in the case of Jeddah, Saudi Arabia. They found that no interventions against the monkeypox would leading to the most life loss and the long-term worst effect on GDP. Quarantine facilities would decrease the death and economic crisis caused by monkeypox. Vaccination would effectively limit the infection of monkeypox virus and protect the national economy, thus is the best intervention. 

The manuscript discusses the ongoing outbreak of monkeypox which is attractive to researchers in this field regarding the public health. The study used limit parameters and the model is straightforward. Even though they did not use real world data of monkeypox infection, the simulation is scientific and finally provides a reference for people to make policies. There is little to criticize, only few remarks and suggestions for improvement:

-Although the study simulates the monkeypox virus spread, the authors should discuss the real-world infection data and try connecting their model to natural infection in human society.

-Some spelling checks: line 62 counties-countries; line 66 monkey pox vs line 67 monkeypox.

Reviewer 2 Report

Summary

The author sought to better understand interventions to mitigate monkeypox spread, including no interventions, quarantine and vaccination. In the long term, the author suggests vaccination would provide the most benefits as compared to the other interventions.

Strengths

·         The authors applied sound logic and statistics to correlate cost vs benefits of interventions to mitigate the monkeypox outbreak.

Weaknesses

·         The authors cite a study that investigated the spread of respiratory infections throughout a population and determined people make an average of 8 contacts per day on average. Here, the authors used this statistic as a basis for the study on monkeypox spread. Two major concerns with using this statistic for monkeypox spread; 1) monkeypox can transmit by skin-to-skin contact and via surface transmission in addition to respiratory spread. The authors should elaborate if this citation looked specifically at respiratory virus that can also spread by skin-to-skin and surface transmission. 2) Did this Leung et al study investigate viral spread across a highly populated city, such as Jeddah with 4.8 million people? 8 contacts per day seems low for a highly populated city. Are there other studies investigating viral spread in a large population? The author could summarize additional studies and use an average number of contacts per day for this analysis.

·         The introduction would benefit with a more in depth explanation of how monkeypox is transmitted with citations, and compared to how other respiratory viruses are transmitted.

·         Monkeypox should be consistently spelled the same way throughout the manuscript, for example lines 29 and 39 state Monkeypox, while other lines state monkeypox and the title states Monkey-pox. Line 66 “monk pox”.

·         Line 265 incorrectly states “…the 4.8 residents of Jeddah”. This sentences should probably state 4.8 million residents.

Overall, this manuscript lacks significant depth and needs to be improved.

Reviewer 3 Report

  Attached  you  can  find  my  comments !

Round 2

Reviewer 2 Report

The author has sufficiently added important citations and references to the introduction regarding monkeypox transmission and other viral outbreaks. However, the study inadequately explained the rationale for the basis of the simulation. Particularly the study did not properly address my previous comment of "Are there other studies investigating viral spread in a large population? The author could summarize additional studies and use an average number of contacts per day for this analysis." 

Author Response

Thank you for the review and the feedback on what could go on improving the design section of this study. As you requested, I summarized 3 additional research studies investigating social conduct behavior patterns among human populations during spread of different diseases. From these additional studies, the average number of contacts remained within a minimum of 4.9 persons and 10.4 persons, presenting an average of about 8.75 contact per person per day, and thus justifying the use of the 8 contacts per day per person as I had earlier adopted.

In the design section, I added a statement to improve it by indicating that the simulation using an average of 8 contacts per day per person is supported by summaries and averages from across different studies on physical contact persons across different populations during viral pandemics.

Thank you.

Reviewer 3 Report

Attached ,  you  can  find  my  comments !

Author Response

Thank you for the review and the comments. I did proofread the entire manuscript and made minor adjustments on grammar and writing mechanics as I thought necessary to improve the readability of the study.

Thank you.